# Hugs and Cortisol Awakening Response the Next Day: An Ecological Momentary Assessment Study

**DOI:** 10.3390/ijerph20075340

**Published:** 2023-03-30

**Authors:** Chelsea E. Romney, Amber Carmen Arroyo, Theodore F. Robles, Matthew J. Zawadzki

**Affiliations:** 1Department of Psychology, Brigham Young University, Provo, UT 84602, USA; 2Department of Psychological Sciences, University of California Merced, Merced, CA 95343, USA; 3Department of Psychology, University of California Los Angeles, Los Angeles, CA 90095, USA

**Keywords:** cortisol awakening response, affectionate touch, hugging, ecological momentary assessment

## Abstract

Previous research suggests that affectionate touch such as hugs might downregulate stress systems such as the hypothalamic pituitary adrenal (HPA) axis. However, the current literature lacks in generalizability beyond the laboratory setting and outside the context of romantic relationships. The cortisol awakening response (CAR) is a measure of the HPA axis and is responsive to daily fluctuations in stress and social information. However, associations between affectionate touch and the CAR have never been assessed. This study used ecological momentary assessment (EMA) to measure daily hugging behaviors in 104 first-year college students and salivary cortisol to assess the CAR. Participants who reported more daily hugs in their social interactions had significantly smaller CARs the next morning compared to days they reported fewer hugs. This study contributes to the literature on social interactions and stress responsive systems and emphasizes the importance of assessing affectionate touch behaviors such as hugs that can be exchanged outside the context of romantic relationships.

## 1. Introduction

Affectionate touch such as hugs can be used in social settings to communicate greetings, excitement, support, and friendship. From Harry Harlow’s (1965) seminal work with rhesus monkeys, we know that a lack of contact comfort may result in major issues with psychological and physical development [1]. In the recent context of social distancing in response to the COVID-19 pandemic, the study of in-person affectionate touch is timely, especially as many of us are re-entering the social space and deciding whether it is safe to engage in affectionate touch behaviors.

Affectionate touch is physical touch interactions that are intended to demonstrate positivity toward another person (e.g., love, care, fondness, and appreciation) [2]. Jakubiak and Feeney (2016) posit a model in which affectionate touch is processed through relational–cognitive and neurobiological pathways to impact relational, psychological, and physical well-being [3]. They assert that stress responses are influenced by affectionate touch receipt. Laboratory research suggests that affectionate touch may blunt the hypothalamic–pituitary–adrenal (HPA) axis response to stress that produces the hormone cortisol. In a study of couples in the laboratory, women who received affectionate touch through neck and shoulder massage before a stressor had significantly lower cortisol and other physiological stress responses, such as heart rate, to the Trier Social Stress Test (TSST), compared to women who received verbal social support or no social interaction with their partner [4]. In another TSST study in couples, both men and women had lower physiological stress response markers (i.e., lower cardiovascular reactivity) when they had a 10 min period of handholding and a 20 s hug before the stressor, compared to people who received no partner interaction [5].

The stress-buffering effect of affectionate touch has also been studied outside of laboratory settings in daily life. In a study of daily partner affectionate touch, more affectionate touch was associated with lower overall cortisol output, measured as the area under the curve (AUC). In this study, married women who reported more instances of affectionate touch also had lower overall levels of salivary cortisol compared to the days they reported fewer instances of affectionate touch [6].

In traditional stress theory, stressors are viewed as threats in the environment that lead to activation in stress-related physiological systems, including the HPA axis [7]. Yet, a more recent stress theory—the Generalized Unsafety Theory of Stress (GUTS) [8]—proposes that safety cues are important when understanding stress perceptions and reactions. GUTS argues that people operate in a default mode of stress responding but can deactivate that response when the environment is perceived as safe, which primarily happens in social contexts. The complementary Social Baseline Theory posits that humans evolved as part of a social context and thus are primed to both interact and seek safety with other humans [9]. More so, it argues that non-social moments may be perceived as stressful due to more effort one must exert to survive when alone, and thus people have a drive to seek social connection, including through social contact. Both these theories argue that social-based safety signals are wired into our functioning, meaning they are largely perceived unconsciously and involve perseverative cognition. For example, a study in everyday life revealed that hours in which one reported being with their partner predicted lower arousal levels, as indexed through electrodermal activity, than when one was separated, an effect that was out of the conscious awareness of the participants [10]. As such, these theories offer a mechanism for how a positive experience at one time may continue to exert positive effects later.

Since affectionate touch has been tied to lower cortisol responses to stressors and challenges, we hypothesize that an environment in which social interactions include hugs may act as a safety signal for a person and result in lower physiological arousal while in those environments. Cortisol release follows a diurnal pattern in which levels decrease throughout the day, reach their lowest levels overnight, and then increase rapidly just before and during awakening. The increase during morning awakening is called the cortisol awakening response (CAR). The CAR may represent attempts to mobilize energy stores that the brain anticipates needing to cope with the stressors and challenges of the day [11,12,13,14]. Yet, too much cortisol may be mobilized under stressful conditions as one “over-prepares” for the day (for example, this might occur if there have not been recent cues of safety). The CAR may also be related to mental health outcomes, with larger CARs predicting higher risks of depressive disorder in young adults one year later [15]. Regarding physical health, young adults with a history of acute or chronic illness show larger CARs compared to young adults with no history of illness [16]. Larger CARs have also been associated with multiple social predictors including higher self-reported loneliness, and smaller CARs have been associated with more positive social interactions from the previous day, such as giving and receiving help [17,18,19].

The current study is the first to examine the association between affectionate touch, expressed through hugging, and the CAR in daily life. Given evidence that more positive social environments may result in smaller CARs (i.e., CARs that do not over-prepare for the day) and that the CAR may be influenced by the events of the previous day [20], we posited that environments in which social interactions included more hugging would signal a safety cue and thus predict smaller CARs the next day (as suggested by work showing that the prior night’s stress reports affects the next morning’s appraisals of stress) [21]. We utilized ecological momentary assessment (EMA) to examine these associations as it is a sampling method that measures a behavior (such as affectionate touch) repeatedly throughout the day and often close to the time the behavior occurred. EMA overcomes two primary limitations seen in previous affectionate touch research including the limited external validity of laboratory settings and recall bias in daily diary studies. Based on the studies of affectionate touch in romantic partners, we hypothesized that more hugs during social interactions over one day would result in smaller CARs the next day, indicating that the anticipatory energetic burden for the next day is smaller when affectionate touch is received throughout the day.

## 2. Materials and Methods

### 2.1. Participants

This study’s sample was taken from a larger study that recruited University of California students living in on-campus housing at the beginning of the first year of college in 2018. Participants in the current study were University of California Merced students living on campus who elected to provide saliva samples as part of the greater study and were at least 18 years of age. Most of the sample identified as Mexican American (50.4%) or other Latine (12.4%). The sample was 54% female, and the participants were either 18 years old (85.7%) or 19 years old (14.3%). Since participants were adolescents, socioeconomic status (SES) was measured through participants’ childhood housing situation. Participants parents’ housing situation was mostly unstable (64.6%), and most participants were first-generation college students (64.3%) (Table 1).

### 2.2. Procedure

Participants were recruited through on-campus flyers and the incoming University of California student Facebook group. Participants attended an initial laboratory session where they provided their informed consent and received instructions on the study procedures. Hugging was reported via text-messaged EMA, and cortisol samples were collected at home by participants. Hugging EMAs and salivary cortisol were collected during the same week on the same days of the week (Tuesday–Thursday) for all participants, during the second full week of classes. Participants were provided insulated bags for the temporary storage of cortisol samples. In the bag to return saliva samples to researchers, participants completed a questionnaire regarding their compliance to the sampling procedure, timing, and behaviors they were instructed to refrain from. Participants also responded with Yes or No when prompted about engaging in eating, drinking, smoking, and exercising between the two morning samples and 12 h before the first morning sample. At the completion of the study days, participants were paid USD 25 to their on-campus student accounts.

#### Ecological Momentary Assessment Sampling

EMAs were sent to participants five times per day using a stratified sampling protocol. Each day, moments were randomly sampled within a 3 h block of time to sample the entire waking day and to ensure that the moments sampled within blocks were random. The earliest 3 h block of time began at 8:00 am, and the latest 3 h block ended at 11:00 pm. Participants were instructed to complete the EMA measures as soon as they received them. Responses made 30 min after the prompt time may have introduced bias because participants may have been providing recollections at convenient, and thus not random, times. Thus, EMA surveys were closed 30 min after they were sent so participants would not be able to complete late responses. The EMA surveys were administered in Qualtrics and were sent as internet hyperlinks via text message to participants’ cellular devices. An online service for sending group text messages in academic settings (Remind, n.d.) was used to send the link to the survey directly to participants.

### 2.3. Materials

#### 2.3.1. Hugging

Participants were asked if they had hugged anyone since their last EMA prompt. Participants only received this question if they reported that they had had a social interaction since the last prompt. The response scale was Yes (1) or No (0). Those values were summed for each day, divided by the number of prompts the participant completed that day, and multiplied by 100. Thus, these scores indicated the percentage of reported interactions for which a hug occurred during that interaction for each sampling day.

#### 2.3.2. Cortisol

Saliva was collected using an absorbent swab (SalivaBio Oral Swab from Salimetrics, Carlsbad, CA, USA). Participants were given written and verbal instructions on proper sampling methods and handling of the absorbent swab. Participants were instructed to avoid touching the cotton swab with their fingers. For waking samples, participants were instructed to place the cotton swab under their tongue for two minutes immediately upon awakening and before getting out of bed. After the two minutes, participants were instructed to return the cotton swab to the Salimetrics collection tube, using their mouth and not their hands, and closed the tube with the cap. Then, the samples were placed in a plastic bag. To ensure the effectiveness of the instructions, participants completed a practice sample in the laboratory with the researchers to troubleshoot any issues and answer questions regarding the participants’ self-sampling method. Participants were also sent an instructional video via SMS the day before sample collection that repeated all sampling instructions.

Participants were instructed to collect the first sample immediately upon waking and to collect the second sample 30 min after waking, according to expert guidelines [22]. After both the waking sample and the 30 min post-waking sample for the day they were collected, participants were instructed to immediately bring the plastic bag with both samples to the saliva sample drop-off stations located outside of their dormitory building. Upon arrival, samples were immediately placed into a cooler with ice packs and returned to a −20° freezer within an hour of being returned to the saliva sample drop-off station. Salivary cortisol was stored and assayed at the UC Merced Psychoneuroendocrinology laboratory. Samples were stored in a −20° freezer until the time of assay. The samples were assayed with a chemi-luminescence immunoassay technique using an assay kit (Salimetrics Assay #1-3002 Kit, Carlsbad, CA, USA). The lower limit of detection was <0.003 ng/mol (intra-assay CV = 0% to 118%; inter-assay CV = 8.14%). Based on the Salimetrics (Appendix A) recommendations for checking sample reliability, there were no values in the sample that met criteria for outliers.

### 2.4. Overview of Analyses

Intercorrelations for the percentage of interactions including hugging and EMA prompt completion for each day were conducted to assess if the number of hugs reported was related to compliance to sample completion. Correlations between cortisol values and sampling times for each of the sampling days were conducted to assess if cortisol release followed the expected pattern, based on expert guidelines [23] (Table A1). Intercorrelations were assessed between the study variables. For the research question, two models were tested. The first model was tested with the outcome (next-day CAR) and the predictor (previous-day hugs). The second model tested the same relationship with the inclusion of covariates.

#### 2.4.1. Hugging

Hugging was assessed at both the within-person daily level and at an aggregated between-person level. Daily hugs_wp_ was the percentages of reported social interactions that included hugs each day, centered around the participant’s average percentage of social interactions with hugs for all three sampling days. Thus, variable hugs_wp_ referred to a person’s deviation from their average social interaction with hugs percentage each day. Hugs_bp_ was the average percentage of social interactions with hugs across all three sampling days.

#### 2.4.2. Cortisol Awakening Response and Lagged Effects

CAR was calculated by subtracting the waking sample from the 30 min post-awakening sample for each of the two days, resulting in a unique CAR for each day. Hugging scores (EMA aggregates as described above) were obtained the day before each CAR was obtained. Participants’ CAR was predicted from the hugging score from the day before to reflect the influence of the functioning the previous day. The CAR values represented were natural log transformed.

#### 2.4.3. Covariates

Participants’ sex was included as a covariate due to the consistent finding in the literature that women exhibit a larger and more prolonged CAR than men [23,24,25,26].

Participants self-reported on day-to-day factors that may influence cortisol. Since cortisol values start to rise in the morning following a diurnal rhythm whether someone is awake yet or not, participants’ waking time was accounted for in analyses. The average waking time in the sample was 7:18 am on Day 1 and 7:45 am on Day 2. In the analyses, the waking time covariate was centered at 7:00 am for both days. Participants were instructed to complete their second sample 30 min after waking and collecting their first sample. Participants self-reported the timing of their samples and were considered non-compliant if they collected their 30 min post-waking sample more than 10 min earlier or later than 30 min after their self-reported waking time. A dummy variable was created that indicated whether a participant was compliant (0) or non-compliant (1) to the sample timing. There were two participants with non-compliant samples on Day 1 and five participants with non-compliant samples on Day 2.

#### 2.4.4. Data Analyses

To account for the hierarchically nested structure of the data (measurement occasions *t*, nested within persons *i*), a multi-level model was used in HLM (Version 7). The first level of the model included the within-persons variables (i.e., variables collected at the daily level). The second level of the model included the individual or “between-subjects” characteristics (i.e., variables meant to assess personal characteristics, instead of daily variation). Two models were tested, predicting CAR the next day. The first model (1) only included the key predictor, daily hugs_wp_.
(1)$$\begin{array}{l} \text{Level-}1\;\mathrm{Model}\\ \;\text{Next-day}\;{\mathrm{CAR}}_{\mathrm{ti}}=\pi_{0i}+\pi_{1i}(\mathrm{daily}\;{\mathrm{hugs}}_{\mathrm{wp}.(t-1)i})+e_{\mathrm{ti}}\\ \text{Level-}2\;\mathrm{Model}\\ \;\pi_{0i}=\beta_{00}+u_{0i}\\ \;\pi_{1i}=\beta_{10} \end{array}$$


The second model (2) was identical to the first model with the inclusion of potential confounds and covariates to test if the effect remained. Random intercepts but no random effects were calculated in the model. The coefficients estimated in both models were the final estimation of fixed effects with robust standard errors.
(2)$$\begin{array}{l} \text{Level-}1\;\mathrm{Model}\\ \;\text{Next-day}\;{\mathrm{CAR}}_{\mathrm{ti}}=\pi_{0i}+\pi_{1i}(\mathrm{daily}\;{\mathrm{hugs}}_{\mathrm{wp}.(t-1)i})+\pi_{2i}({\mathrm{compliance}}_{\mathrm{ti}})+\pi_{3i}({\text{waking time}}_{\mathrm{ti}})+e_{\mathrm{ti}}\\ \text{Level-}2\;\mathrm{Model}\\ \;\pi_{0i}=\beta_{00}+\beta_{01}({\mathrm{sex}}_{i})+\beta_{02}({\mathrm{hugs}}_{\mathrm{bp}.i})+\beta_{03}({\mathrm{ethnicity}}_{i})+u_{0i}\\ \;\pi_{1i}=\beta_{10}\\ \;\pi_{2i}=\beta_{20}\\ \;\pi_{3i}=\beta_{30} \end{array}$$


## 3. Results

### 3.1. Preliminary Analyses: Hugging and EMA Prompt Completion

Descriptive statistics and intercorrelations for hugging and EMA prompt completion for each day were completed. Over the three sampling days, participants completed, on average, three out of five prompts per day (M = 8.98, SD = 3.8, range = 1–15). On average, they reported social interactions in 81% of their prompts (M = 7.31, SD = 4.2, range = 0–15), and hugging occurred in 15% of these reports (M = 1.35, SD = 2.2, range = 0–12).

To look at whether there were systematic biases in reports of hugging and completion, we examined the raw totals of hugging reports (as the percentage conflates hugging totals and number of reports/missingness). Higher raw reports of hugs on each of the days were related to higher reports of hugs on the other days (correlation range = 0.55 to 0.65). The amount of EMA prompts completed was not correlated with the raw number of hugs reported on any of the three days (correlation range = 0.002 to 0.02), suggesting that compliance with sample completion was unrelated to the proportion of hugs reported. Raw hugging amounts were modeled as a function of study day to assess if hugs were different on any of the study days. The results determined that the raw numbers of hugs were not predicted by study day (unstandardized coefficient = −0.004, standard error = 0.007, *p* = 0.57).

### 3.2. Preliminary Analyses: Cortisol and Potential Confounding Variables

Descriptive statistics and intercorrelations for cortisol values and sampling times for each of the sampling days were calculated. Higher cortisol concentrations at both waking and 30 min post-waking were related to higher cortisol concentrations both on the same day and between days (correlation range: 0.5 to 0.36). Cortisol values followed the expected pattern (waking sample day 1: M = 0.279 ng/mol; waking sample day 2: M = 0.299 ng/mol), with about a 50% increase in the first 30 min after awakening (Day 1: M = 0.480 ng/mol; Day 2: M = 0.455 ng/mol). Higher daily waking cortisol values were associated with smaller CARs on both days (correlation range: −0.42, −0.22) (see Appendix B for all values).

### 3.3. Preliminary Analyses: Covariates, Independent, and Dependent Variables

Table 2 displays the descriptive statistics and the correlations between the variables and covariates used in the models. Hugs_wp_ represents the person-centered value for the percentage of social interactions with hugs each day or the deviation from a person’s average percentage of social interactions with hugs. Hugs_bp_ is the overall average percent of social interactions with hugs reported out of all the EMA samples completed.

Hugs_wp_ each day were significantly correlated, with higher deviations from an individual’s average on one day predicting lower deviations on the other day (*r* (103) = −0.56, *p* < 0.001). Hugs_wp_ were not significantly correlated with hugs_bp_ on any of the days. There was no evidence of a correlation with compliance or sex.

The two daily CAR values were not significantly correlated. CARs were not correlated with compliance, medication use, ethnicity, or SES. However, CARs on Day 1 were significantly associated with participant sex, with females (M = 1.66, SD = 0.05) displaying larger CARs than males (M = 1.63, SD = 0.034), *F* (2, 105) = 10.76, *p* = 0.001. There were no significant associations between CAR and hugs_wp_ on the previous day, same day, or the following day (correlation range = 0.111 to 0.082).

### 3.4. Hugs and CAR

The analyses of the two models are detailed in Table 3. The first model examined the association between the percentage of social interactions with hugs and CAR the next day. CAR the next day was modeled as a function of the daily deviation from participants’ own average hug percentage for all the study days (daily hugs_wp_) and a residual component. As hypothesized, participants displayed lower CARs following days where they reported more hugs during social interactions compared to their average number of social interactions with hugs for all the sampling days (unstandardized coefficient = −0.0005, SE = 0.0002, *p* = 0.011).

The second model included all covariates and potential confounds, including sex and the participant’s average percentage of hugs for all the sampling days. Level 1 contained the daily compliance variable and the participant’s daily waking time. Level 2 added ethnicity. None of these added variables were associated with CAR. Sex and daily hugs_wp_ predicted CAR the next day. Females displayed larger next-day CARs (unstandardized coefficient = 0.02, SE = 0.007, *p =* 0.004), and daily hugs_wp_ significantly predicted lower next-day CARs (unstandardized coefficient = −0.0004, SE = 0.0002, *p* = 0.034).

To compute an effect size for the association between hugs and next-day CARs, we used the r2mlm package for R that can provide R^2^ and ΔR^2^ estimates for multilevel models under a recently developed framework [27]. We compared a model with covariates only to a model with covariates and hugs_bp_ and daily hugs_wp._ In this framework, the difference in proportion of total outcome variance was explained by the level-1 predictor daily hugs_wp_ $\Delta R_{t}^{2\left(f1\right)}$ = 0.08. The difference in the proportion of within-cluster (within-person) outcome variance was explained by daily hugs_wp_ $\Delta R_{w}^{2\left(f1\right)}$ = 0.11. To date, R^2^ metrics for multilevel models are a very new development, and thus, benchmarks for the effect size magnitude in multilevel modeling have not been developed.

## 4. Discussion

This study is the first to assess associations between hugging and CAR as a marker of HPA axis function in daily life in a diverse first-year college student sample. Compared to days when participants reported lower percentages of social interactions with hugs, on days when participants reported higher hug percentages, they showed lower CARs the following day. These findings are in line with previous laboratory and daily diary studies of HPA axis patterns and affectionate touch [4,6,28], such that increased affectionate touch is associated with reduced daily cortisol secretion and reduced HPA axis response to stressors.

Our findings complement and extend past research on the effects of affectionate touch on stress responses in the laboratory [4,29] and in daily life [6]. We found smaller CARs were associated with more social interactions with hugging the day before, but there was no evidence of an association with a person’s average levels of social interactions with hugging for the duration of the study. This is in line with previous research on married couples that found day-to-day variations in daily affectionate touch, but not overall mean intimacy levels, were associated with reduced salivary cortisol secretion [6]. This study also extends the research on affectionate touch behavior by measuring affectionate touch as hugs, rather than behaviors such as kissing and handholding [4,6] that may be limited to the romantic relationship context. While affectionate touch behaviors can vary across cultures [30], ethnicity did significantly correlate with the CAR effect, suggesting our findings may hold across ethnicities and cultural backgrounds.

The mechanism for the effect of previous day social functioning, including hugging during social interactions, and CAR is not currently known. However, some studies suggest that the CAR is influenced by sleep, with poorer sleep related to larger CARs the next day [31,32]. The anticipation of stress the following day may also influence the CAR, with more anticipatory stress about the next day predicting increased CARs the next day [33]. These effects could be explained by the GUTS model [8] and Social Baseline Theory [9] in that having social interactions with hugging could be a critical safety signal that resulted in no expression of a stress response. In turn, lower arousal during the day may have resulted in better sleep. Additionally, the affectionate touch may have alerted people to additional social support resources available (and subsequent safety signals) to facilitate coping with future stressors.

### Limitations and Future Directions

The primary limitation of this study is that it is lacking the social context in which the hugs were received and reported. Indeed, past studies have found that CAR may be particularly responsive to social information. For example, in adolescents, prosocial behaviors with friends are related to lower CARs the next day [19]. Similarly, reports of providing support to family members also predict lower CARs the next day [18]. While we knew the specific affectionate touch behavior (hugging), we did not collect information on who the hugs were exchanged with, who initiated the hugs, and the participant’s subjective meaning of the hug. To understand if hugs are a receipt of social support, the provision of social support, or a mutual exchange, future studies should collect information on the full context of the exchange. Additionally, it may be important to understand when and in what context the hugs occurred. Experiences of support, including hugs, early in the morning may have indirect effects on CAR the next day by buffering the stress experiences, promoting coping responses and faster recovery from stress, and/or disrupting engagement in perseverative cognitions. EMA methods such as those used in the current study would provide a sufficient method for accomplishing this by including prompts about who with and why the hugs took place.

Although the focus of the current study was hugging, a review of the literature suggests that multiple acts of affectionate touch may influence stress responsive systems. For example, affectionate touch in romantic couples, including massages, may downregulate multiple stress systems [29]. While there is some precedence for using hugs specifically as an easily reported measure of affectionate touch [34], future studies should allow for reports of all affectionate touch behaviors, which may vary between people and in the context of different social relationships. Future studies should also expand to other age groups and populations outside of adolescents and college students.

Finally, we tested hypotheses on first-year college students as it allowed an opportunity to examine how this sample adjusted to changes to a new environment, including new social connections. We expected this would provide variability in the data as to the range of CARs each morning and the frequency that hugs would be part of social interactions. Yet, the appearance of hugs among these emerging relationships may proxy a different construct than for more established ones. For example, whether these first-year students engaged in hugs may indicate how well their adjustment is faring and their level of social integration with their peers. For more established relationships, a hug might indicate the quality of that relationship at that time or reveal an aspect of the relationship dynamic. Thus, future work should aim to extend these findings on new samples, including those with different relationship types and lengths.

## 5. Conclusions

Overall, these findings complement and build on previous research findings that affectionate touch influences HPA axis functioning by adding the finding that more hugging is associated with a lower cortisol awakening response (CAR) the next day. Affectionate touch may act as a safety signal that is considered when anticipating stress for the next day. This reduction in anticipation of stress may result in a decrease in perceived energetic demand for the next day, reflected in reduced CARs. Future studies should unpack this potential pathway by investigating the effect of affectionate touch on subjective stress measures and anticipatory stress measures. To allow further interpretations about support provided or received, more context on the affectionate touch behavior should be measured, including hugging partners and the subjective meaning of the hug.

## Figures and Tables

**Table 1 ijerph-20-05340-t001:** Sample characteristics.

Variable	Categories	N	%
Sex	Female	61	54.0
	Male	49	43.8
	Unidentified	2	1.8
Age	18 years old	96	85.7
	19 years old	16	14.3
Ethnicity ^1^	Mexican	57	50.4
	Other Latine	14	12.4
	Pacific Islander	10	8.8
	White	6	5.3
	Native American	5	4.4
	Indian	5	4.4
	Other	5	4.4
	Other Southeast Asian	4	3.5
	Chinese	3	2.7
	Middle Eastern	1	0.9
	Black	1	0.9
Socioeconomic Status (Childhood Housing Situation)	Unstable	3	2.7
	Parents rented	53	46.9
	Parents owned then rented	8	7.1
	Parents rented then owned	7	6.2
	Parents owned home	40	35.4
Student Status	First generation	72	64.3
	Non-first generation	39	34.8

Note. ^1^ Not all possible categories for participant ethnicity are included here, only those endorsed by the participants included in this study. The ethnicities that are not represented in this sample are Japanese and Korean.

**Table 2 ijerph-20-05340-t002:** Means, standard deviations, and correlations between variables.

	Categories	%	1.	2.	3.	4.	5.	6.	7.	8.	9.	10.
1. Sex ^1^	Female Male	540.5 430.8	-									
2. Ethnicity ^2^	Latine Non-Latine	620.8 370.2	0.16	-								
3. Day 1 Compliance ^3^	Non-compliant Compliant	10.8 950.5	0.01	0.04	0.04	0.02	-					
4. Day 2 Compliance ^3^	Non-compliant Compliant	50.4 940.2	−0.05	0.09	0.24 *	0.04	−0.03	-				
	**Mean**	**SD**	**1.**	**2.**	**3.**	**4.**	**5.**	**6.**	**7.**	**8.**	**9.**	**10.**
5. Day 1 Hugs_wp_ ^4^	10.99	160.12	0.02	0.12	−0.10	0.06	0.01	−0.15	-			
6. Day 1 CAR ^5^	10.65	0.048	0.31 **	−0.02	−14	0.02	0.01	−0.07	−0.19	-		
7. Day 2 Hugs_wp_ ^4^	00.96	170.10	0.03	−0.20 *	−0.03	−0.27 **	0.01	0.10	−0.56 **	0.15	-	
8. Day 2 CAR ^5^	10.64	0.05	0.06	−0.01	0.13	0.13	0.02	−0.05	0.04	0.18	−0.13	-
9. Hugs_bp_ ^6^	200.24	300.4	0.02	0.06	−0.03	−0.03	−0.14	−0.01	0.04	−0.16	0.10	0.003

Note. ^1^ 0 = male, 1 = female; ^2^ Non-Latine = 0, Latine = 1; ^3^ 0 = compliant, 1 = non-compliant; ^4^ the daily deviation from the average percentage of hugs for all 3 days of sampling (wp = within person); ^5^ values are natural log transformed; ^6^ the percentage of hugs across all the sampling days out of the EMAs they completed for all the sampling days (bp = between person); * *p* < 0.05, ** *p* < 0.01.

**Table 3 ijerph-20-05340-t003:** Hierarchical linear models predicting CAR by hugging percentage using full information maximum likelihood estimation.

Fixed Effects	Daily Hugs_wp_		Daily Hugs_wp_ with Covariates	
	Unstandardized Coefficient	SE	Unstandardized Coefficient	SE
Intercept Next-day Cortisol Awakening Response	1.65 **	0.004	1.64 **	0.012
Daily Hugs_wp_ ^1^	−0.00005 *	0.0002	−0.0004 *	0.0002
Hugs_bp_ ^2^			−0.0005	0.0001
Sex ^3^			0.020 **	0.007
Ethnicity ^4^			−0.0058	0.007
Compliance ^5^			−0.007	0.010
Waking time ^6^			−0.00006	0.00005
Random Effects				
Level 1 Intercept	0.00031 *	0.01768	0.00031 *	0.01773
Residual	0.002	0.045	0.002	0.044

Note: ^1^ the daily deviation from the average percentage of hugs for all 3 days of sampling (wp = within person); ^2^ the percentage of hugs across all the sampling days out of the EMAs they completed for all the sampling days (_bp_ = between person); ^3^ 0 = male, 1 = female; ^4^ Non-Latine = 0, Latine = 1; ^5^ 0= compliant, 1= non-compliant; ^6^ waking time was centered around 7:00 am; * *p* < 0.05, ** *p* < 0.01.

## Data Availability

All data can be provided upon request.

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
