# Peer review of "Hugs and Cortisol Awakening Response the Next Day: An Ecological Momentary Assessment Study"

_ijerph, 2023, doi:10.3390/ijerph20075340_

Round 1
Reviewer 1 Report
The current research was designed to assess whether hugs in daily life predict the cortisol awakening response (consistent with the idea that hugs are a safety signal that downregulates the stress response). This work was interesting and clearly described. My questions and recommendations intended to improve the manuscript are as follows:
- The authors reference the GUTS theory; a related theory they may also consider including is the social baseline theory (e.g., Beckes & Coan, 2011; Coan & Sbarra, 2015). Based on the authors’ descriptions of GUTS, these theoretical perspectives seem to have a great deal of overlap.
- Given that there are cultural differences in the prevalence of affectionate touch, could the authors comment on how their results may generalize to other samples that are not majority Latinx.
- The equations presented show “daily hugs wp.ti” predicting “Next-day CAR ti.” Although the equation says “Next-day” the consistent subscript “t” suggests that hugs and CAR were modeled on the same day. Can the authors clarify their equation to match the description of lagged analyses?
- I have often seen between-person and within-person measures included in the same model. In other words, I would have expected to see within-person deviations in daily hugs predicting CAR controlling for between-person hugs (grand-mean centered). This is the approach recommended by Bolger & Laurenceau (2013) in their intensive longitudinal methods book. Why did the authors opt not to include the between-person measure of hugs in their model? How do the results change with this variable included as a control?
- Relatedly, the authors write in the discussion: “We found smaller CARs were associated with more social interactions with hugging the day before, but no evidence of an association with a persons’ average levels of social interactions with hugging for the duration of the study. This is in line with previous research on married couples that found day-to-day variations in daily affectionate touch, but not overall mean intimacy levels, was associated with reduced salivary cortisol secretion [6].” This quote makes it sound as if they tested whether between person hugging (i.e., overall mean intimacy levels” were associated with CARs; however, the tested links between average levels of social interactions and hugging do not do so. Can you please clarify?
- In lagged analyses, I would have also expected to see the current day’s CAR included as a predictor of next day CAR. Doing so would more directly assess residualized change in CAR that is associated with daily hugs.
- The authors state that they have no information about hugs were exchanged with. Were students living at home (suggesting that their hugs may have been with family members), or were students living on campus (suggesting their hugs were with new friends and acquaintances they had met in the first weeks of the semester)?
- Relatedly, I think the authors should think more about the limitations inherent in using a sample of freshman during their first few weeks of the semester. This is likely a particularly stressful time in which “over-preparing” for the next day might even be an appropriate stress response. Moreover, hugs in this new context could be a metric of something like social integration versus isolation, unlike in other contexts where hugs could be a metric of relationship quality in existing relationships.
- Finally, it would be helpful if the authors indicate or conceptualize the size of their effects. How impactful is hugging? The estimates are small but are from unstandardized models.
Reviewer 2 Report
The manuscript with the title “Hugs and Cortisol Awakening Response the Next Day: An Eco-2 logical Momentary Assessment Study” (ijerph-2224146) is well written.
The introduction has a clear structure, derived research questions are appropriate. Methods including study design and analyses as well as results are sound and clearly presented.
Discussion including limitations and conclusion are also clearly presented and its length is appropriate.
In the following I have some comments / suggestions:
(1) I do not think that the participants’ childhood housing situation does present the SES of the person well. By dichotomization its validity is even more diminished.
(2) I do not think that the consideration of taking medication makes sense because only three participants reported taking medications on study days at all (and the substances are very different moreover).
(3) Can you provide more information towards effect sizes of the results?
(4) Higher raw reports of hugs on each of the days were related to higher reports of hugs on the other days (correlation range= .55 to .65). -> in my opinion this might be a hint towards differences between the participants, so that differences in the participants (for instance in terms of social integration) might be more important in terms of CAR rather than the hugs.
(5) In the discussion you recommend future studies should collect information on the full context of the exchange of hugs, which seems quite important. Further studies could also track stressors on the day before or the same day which influence CAR.
(6) A further limitation is the limited generalizability in terms of age (only adolescences) and status (only students)
Reviewer 3 Report
The manuscript and the research presented in it should be considered interesting. I have only minor comments and questions, which I present below.
There is some ambiguity about the Childhood Housing Situation variable. I am interested in why it was chosen as the SES measure. It is not clear why all situations in which there has been a change (e.g. parents first rented and then owned their own house) are considered unstable. The text shows that the Unstable category includes: Parents rented, Parents owned then rented and Parents rented then owned, while Table 1 has a separate category Unstable. The percentages in the Unstable category do not add up to 64.6%, and all highlighted categories for the Socioeconomic Status (Childhood Housing Situation) variable do not add up to 100%.
It would be interesting to know what percentage of people reported no hugs during the day. Do the authors have any information about the situation in which the hugs appeared?
Round 2
Reviewer 1 Report
I was reviewer 1 on the initial submission. The revision did an excellent job of addressing my earlier concerns.